# The effect of experiential learning interventions on physical activity outcomes in children: A systematic review

Sumantla D. Varman[1,¤,*], Rachel A. Jones[2,3], Bridget Kelly[1], Megan L. Hammersley[1,3‡], Anne-Maree Parrish[2‡], Rebecca Stanley[2‡], Dylan P. Cliff[2,3]

**1** Faculty of the Arts, Social Sciences and Humanities, Early Start, School of Health and Society, University of Wollongong, Wollongong, NSW, Australia, **2** Faculty of the Arts, Social Sciences and Humanities, Early Start, School of Education, University of Wollongong, Wollongong, NSW, Australia, **3** Illawarra Health and Medical Research Institute, Wollongong, NSW, Australia

☯ These authors contributed equally to this work.
¤ Current address: Department of Applied Sciences, College of Engineering Science and Technology, Fiji National University, Natabua Campus, Fiji
‡ These authors also contributed equally to this work
* sdv656@uowmail.edu.au

**Data Availability Statement:** All relevant data are within the paper and its Supporting Information files.

## Abstract

### Background

This systematic review examined the effectiveness of experiential learning interventions for improving children's physical activity knowledge, attitudes, and behaviours. It also aimed to identify intervention characteristics that resulted in the greatest impact.

### Methods

Four databases: Education Research Complete, Scopus, Web of Science and PsychINFO were searched from database inception to January 2023. Eligible studies: (1) included children 0–12 years; (2) assessed the effect of physical activity outcomes on children's physical activity knowledge, attitudes or behaviour and (3) were randomised controlled trials conducted in any setting. Study risk of bias was assessed by two independent reviewers using the Cochrane risk of bias tool. Intervention approaches were categorised, and effect sizes were compared across studies for each outcome.

### Results

Twelve studies were included in the review: ten in school age and two in below five years. For behavioural outcomes, six of eight studies showed medium to large effects (effects size (ES) range: 0.3–0.9), two of the three studies that assessed attitudinal outcomes displayed medium effects (ES range: 0.4–0.5) and both studies that assessed knowledge outcomes displayed medium to large effects (ES range: 0.4–1.3). The two experiential learning interventions among children < 5 years demonstrated small to medium effects on behaviour change (ES range: 0.2–0.5). Effective interventions combined enjoyable practical activities (fitness activities, games and challenges), with behaviour change techniques (goal setting,

**Funding:** The author(s) received no specific funding for this work.

**Competing interests:** The authors have declared that no competing interests exist.

and self-monitoring), were underpinned by a behaviour change theory, and were often of short duration (< 4 months) but intense (several sessions/week). Moderate to high statistical heterogeneity was observed for behaviour outcomes and risk of bias across studies was generally high.

## Conclusions

This review provides some evidence supporting the effectiveness of experiential learning interventions in improving physical activity outcomes in school-aged children. Additional evidence is needed in children <5 years old. Future experiential learning interventions need to strengthen the evidence with rigorous methodological quality and clear reporting of the experiential learning components.

## Introduction

Physical activity is an important component of a healthy lifestyle for children that impacts life-long health [1]. Regular physical activity promotes healthy weight among children [2] and reduces the risk of adverse health outcomes including chronic diseases [3–5]. In recognition of the health benefits of regular physical activity for children, many countries have established national physical activity guidelines for early childhood [6–8] and for school-aged children and adolescents [8–10]. The World Health Organization has also developed physical activity guidelines for children, which recommend at least 60 minutes of moderate to vigorous-intensity physical activity for children aged 5–17 years with the inclusion of vigorous and resistance activities at least three times a week [11], and at least 180 minutes in a variety of types and intensities of physical activities per day for children 1–5 years of age [12].

Globally, less than half of children under 5 years and 5–12 years meet physical activity recommendations [13,14], and this declines with age [15]. Studies show that physical activity levels track from early childhood to adolescence and adulthood [16,17], thus early life is a pivotal time for physical activity promotion. It has been suggested that enhancing children's knowledge about the benefits of and fostering positive attitudes towards physical activity may enable them to change their habits and encourage them to adopt and maintain healthy and active lifestyles [18,19]. A study by Nelson et al. [20] suggests that attitudes toward physical activity may have an impact on activity levels in children and recommends further investigations involving attitudes, self-efficacy, and physical activity. One potentially positive and innovative approach to influence children's physical activity knowledge and attitudes and eventually their physical activity behaviours is experiential learning.

Experiential learning is a learning process in which knowledge is gained through experiences (i.e., hands-on and practical experiences in addition to theory-based components). Experiential learning facilitates learners to create knowledge, skills, and attitudes, and by integrating the experience and perceptions, prompts behaviour change [21]. The core of experiential learning is David Kolb's four-stage learning model: (i) concrete experience, in which learners participate in concrete experiences such as hands-on activities, (ii) reflective observation in which learners develop perspectives e.g., reflections of the experiences, (iii) abstract conceptualization whereby learners identify the significance of the learning experience/knowledge and from which new behaviours emerge, and (iv) active experimentation which is the practical application of the new concept [22]. Reflection is the key component of experiential

learning that enables learners to relate knowledge and behaviour in a systematic way [23]. Experiential learning exposes children to experiences where they can explore, play and become familiar with the materials and concepts that could lead to lifelong learning of the concept [24,25].

Experiential learning physical activity interventions have been implemented in a variety of settings but primarily in educational settings (including pre-formal school settings [e.g., early childhood education and care settings] and schools). These interventions utilised a breadth of experiential learning approaches. For example, Ourda et al. [26] used creative and theatrical activities to measure children's physical activity-related knowledge, Chen et al. [27] used fitness and challenges to measure children's attitudes towards physical activity and Eather et al [28] used modified sports and games to measure physical activity behaviour in children. Outcomes from these studies have been varied with mixed findings in relation to which type of experiential learning activities can increase physical activity in children and how. It is, therefore, crucial to look at the available evidence to understand the effectiveness of experiential learning physical activity interventions among children.

To our knowledge, no systematic review has investigated the effectiveness of experiential learning interventions on children's physical activity-related knowledge, attitudes, and behaviours. A comprehensive review is important given the diversity and type of interventions. Therefore, the purpose of this review was twofold: (1) to investigate the effectiveness of experiential learning interventions for improving children's (birth to 12 years) physical activity knowledge, attitudes, and behaviours; and (2) to identify the characteristics and components of physical activity experiential learning interventions, and the factors surrounding the implementation of the activities that lead to the greatest impact.

## Methods

This review was registered with PROSPERO (no. CRD42018103629) and conducted following Preferred Reporting Items for Systematic Review and Meta-Analysis (PRISMA) guidelines for systematic reviews [29].

### Search strategy and eligibility criteria

Four databases were searched for eligible studies: Education Research Complete, Scopus, Web of Science and PsychINFO. No limits were applied to the publication date and the search was conducted to obtain articles published from database inception to 15 January 2023. Reference lists of included studies were also searched. Included studies were randomised controlled trials (RCTs) or cluster/group RCTs reported in original, peer-reviewed articles. The inclusion of only RCTS was a post hoc protocol deviation as these were deemed as the most robust level of evidence. Studies were excluded if they were not published in English, or were reviews, opinions or descriptive articles. Additionally, in the review registration, both healthy eating and physical activity outcomes are stated but this paper only reports physical activity-related outcomes as healthy eating outcomes have been reported elsewhere [30]. Eligible studies and hence the search terms (Table 1) were guided by the Population, Intervention, Comparison, Outcomes (PICO) framework:

**Population.** In this review, children were defined as those aged between birth to the age of 12 years old. Studies were included if the mean age of participants was between birth and 12 years and excluded if participants were above 12 years at baseline.

**Intervention.** Studies were included where the intervention incorporated an experiential learning activity, with one or more of the following characteristics (1) children played a central role in the activity, allowing them to engage with, and explore, the phenomena; (2) the activity

**Table 1. Search terms.**

| PICO | Booleans | Search terms |
|---|---|---|
| **Population** | AND | "Child*" OR "Preschool" OR "Elementary School" OR "Elementary student" OR "Elementary education" OR "grade 1" OR "grade 2" OR "grade 3" OR "grade 4" OR "grade 5" OR "grade 6" OR "Kindergarten" OR "primary education" OR "primary school" OR "early years" |
| **Intervention** | AND | "Play-based learning" OR "learning through play" OR "experiential learning" OR "Learning cent#red play" OR "Student cent#red learning" OR "Guided play" OR "Facilitated play" OR "Play-based education" OR "Play education" OR "educati* activ*" OR "Interactive learning" OR "Playful pedagogy" OR "Active learning" OR "Experiential education" OR "experience-based learning" OR "program*" OR "intervention" OR "workshop" OR "promotion" OR "project" |
| **Outcome** | AND | "Physical Activ*" OR "fitness" OR "Energy expenditure" OR "physical literacy" OR "Exercise" |

went beyond the provision of information, instruction, encouragement, equipment or change to the environment, i.e., it involved practical components and activities with a focus on a learning task; (3) the activity required children's input and children had to physically do something as part of the learning activity; (4) the children had a level of autonomy in completing some part of the activity that required them to be creative, problem-solve or be reflective; (5) the activity was specifically designed to have a learning experience that enhanced physical activity outcomes, measured post-intervention; (6) the activities had a clear learning task or skill as the outcome; (7) the activity was coordinated/facilitated by a leader such as a teacher or educator, or parent; and (8) the facilitator(s) provided the structure for the activity such as basic instruction, posing questions to invoke problem-solving, creative thinking or reflection [31–33]. The study was excluded if there was no experiential learning activity for children as part of the intervention.

**Setting.** Studies were included from all settings (e.g., school, after-school, preschool, and community).

**Outcome.** A study was included if it had at least one outcome related to physical activity knowledge, attitudes, or behaviour (either self-/parent-proxy reported or objectively assessed). Physical activity was defined as any bodily movement produced by skeletal muscle that requires energy expenditure for example walking, cycling, wheeling, sports, active recreation and play [34].

## Study selection

Study records were imported into EndNote reference software version X9 (Clarivate Analytics, London, UK). Duplicate studies were removed, and two reviewers (M.L.H and GN) independently screened the titles and abstracts. Where discrepancies existed, discussions were conducted between the reviewers to reach a consensus. All studies selected by the reviewers (M.L.H and GN) were then assessed for inclusion by the first author (S.D.V.) at the full-text stage.

## Data extraction

The data extracted included: study authors/year of publication, country of study, study design, theoretical framework, study sample (size, age of participants), intervention (duration, experiential-based learning activities [including theory and practical], outcome measures/tools) and results. The first author (S.D.V.) extracted all data using a devised table, which was checked by a second and third reviewer (D.P.C. and R.A.J.) Disagreement between the reviewers was resolved through discussion with another author (B.K.).

## Quality appraisal

To assess the potential risk of bias in included studies, the revised Cochrane risk-of-bias tool for randomised trials (RoB 2) [35] was independently completed by two reviewers (S.D.V. and Z.Z.), with two additional reviewers (R.A.J. and D.P.C.) consulted if consensus could not be reached. This tool examines five domains: the randomisation process; deviations from the intended interventions (effect of assignment to intervention or effect of adhering to intervention); missing outcome data; measurement of the outcome; and selection of the reported results. The RoB 2 criteria for the overall risk of bias judgement was used [36]. The overall risk of bias was defined using the following criteria: (1) low risk of bias–the study was judged to be at low risk of bias for all domains, (2) some concerns–the study is judged to raise some concerns in at least one domain, but not to be at high risk of bias for any domain (3) high risk of bias–the study was judged to be at high risk of bias in at least one domain or the study was judged to have some concerns for multiple domains in a way that substantially lowered confidence in the result [36].

## Data synthesis and analysis

The experiential learning experiences included a practical component and a theory-based component. To aid the discussion of the results, the practical and the theory components of the interventions were grouped based on the synergies. The practical components were categorized as (1) outdoor and adventure activities; (2) fitness activities and challenges; (3) rhythmic and expressive movement activities; (4) modified sports and games and (5) active play and tummy time. The theory components were categorized as (1) behaviour change and motivational strategies and (2) information and education strategies. A summary of these categories and examples of each category are shown in Table 2.

The effect sizes for the difference between the intervention and control groups on each outcome measure (increased physical activity, improved physical activity attitudes and improved

Table 2. Practical and theory components of experiential learning physical activity interventions.

| Components of experiential learning activities | Categories | Examples |
|---|---|---|
| **Practical** | Outdoor and adventure activities | Hiking, hunting, berry picking, horseback riding |
| | Fitness activities and challenges | Walking, jumping rope, brisk walking, running, fitness challenges. Strength activities using playground equipment. Aerobic exercises |
| | Rhythmic & expressive movement (creative & theatrical) activities | Dancing, yoga, relaxation exercises. Creative gymnastic activities. Movement and theatrical games. Body balancing exercises |
| | Modified sports and games | Active games Playing games at high intensity Indigenous games—double ball game, ring the stick game. Small-sided games Outdoor modified sports |
| | Active play & tummy time | Fun and interactive activities, and prone-based play-child tummy time—child moving arms and legs, playing games on the floor, crawling, hitting a balloon) |
| **Theory** | Behaviour change strategies | Self-efficacy development, self-regulation, and social support. |
| | Motivational strategies | Positive reinforcement, Role modelling. |
| | Reflection | Self-monitoring and reflection (on weekly activities) |
| | Information strategies | Creating awareness (e.g., the importance of tummy time and active play from birth, the importance of being active) |
| | Education strategies | Information sessions regarding physical activity and active lifestyles (e.g., physical education). |

physical activity knowledge) were calculated, regardless of their statistical significance. The pooled standard deviation (SD) was calculated using the following equation from Cohen [37].

$$SD*_{pooled} \sqrt{\frac{(n_1 - 1)SD_1^2 + (n_2 - 1)SD_2^2}{n_1 + n_2 - 2}} \qquad [1]$$

where: $n_1$ is the sample size for group 1 and $n_2$ is the sample size for group 2. The mean difference between the intervention and control groups was divided by the standard deviation (SD) for both groups (pooled SD). Effect sizes were then calculated using Cohen's d [38], formula: $d$ = (M1—M2)/SD pooled. Where M1 is the mean of the intervention group, M2 is the mean of the control group and $SD_P$ is the pooled standard deviation for both groups. Effect sizes were interpreted as small ($< 0.2$), medium ($0.2–0.8$) and large ($> 0.8$) [37]. Intervention approaches for each study were categorised and effect sizes were calculated separately. The intervention approaches and effect sizes were then compared across studies.

Possible causes of heterogeneity in effects were assessed using the $I^2$ statistic, Cochran Q-tests and subgroup analyses. Cochran Q-tests were used to determine the statistical heterogeneity for physical activity behaviour and attitude outcomes, with statistical significance set at $p < 0.1$. All subgroup analyses used a single pooled estimate of Tau-squared (between-study variance). Random effects models were employed to allow for variation between studies. The confidence interval (with 95% confidence limits) was used to show the range of true effect sizes for included studies.

Potential publication bias was assessed by inspecting forest and funnel plots as well as via running Egger's Test, [39], (using a recommended criterion of $p < 0.1$, [40] Additionally, Duval and Tweedie's Trim and Fill method was employed to estimate and impute potential unpublished effects to assess their impact on the pooled effect size and significance [41]. Subgroup analyses could not be performed for knowledge outcomes due to the small number of studies. Additionally, the trim and fill test for physical activity attitude outcome was not run due to not enough valid cases for processing thus no statistics were computed. The calculations for heterogeneity tests, subgroup analysis and publication bias evaluation were performed using the IBM SPSS Statistics 28.0 and the guidance from [42].

## Results

### Study selection

Seventy-eight studies described physical activity-related outcomes in children. Of these, 55 studies did not have a control group and 13 studies were non-randomised controlled trials (NRCTs), thus were excluded. In total, 12 studies were included in the final review, as shown in S1 Fig.

The characteristics of the included studies and their outcomes are summarised in Table 3. Of the 12 included studies, four were RCTs and eight were cluster RCTs (CRCTs). Four studies (4/12) were conducted in the United States while the other studies were conducted in Australia ($n = 3$), Greece ($n = 2$), New Zealand ($n = 1$), Germany ($n = 1$), and Lebanon (n = 1). Most of the interventions were underpinned by the Social Cognitive Theory ($n = 8$) [27,28,43–48], while four studies did not report using any theoretical framework [26,49–51]. Only two studies involved children aged birth—5 years [43,49] and 10 studies involved participants aged 6–12 years.

### Study and intervention characteristics

The majority (7/12) of interventions were conducted in the primary school setting [26,28,45–47,50,51], five were conducted outside of school and in after-school recreation and care

**Table 3. Experiential learning approaches and physical activity outcomes in children.**

| Authors, (Year) Country | Study Design/ Theory used | Sample Size, Age/Grade, Parent Involvement | Intervention (setting, duration, experiential learning practical/theory components) | How experiential learning criteria were met | Measures/tools | Results | Overall Risk of Bias |
|---|---|---|---|---|---|---|---|
| colspan Studies in preschool-aged children (< 6 years old) | | | | | | | |
| (Hewitt et al., 2020) Australia [43] | CRCT SCT | N = 35, mean age 5.9 weeks Yes | Local area health district, parent facilitated. 4 weeks. *Practical component*: weekly tummy time practical (2 hours) mother's group session. *Theory component*: education on the importance, value and benefits of tummy time. Small, achievable weekly goals were set to assist mothers to change tummy time frequency and duration. | • Children played central role. • Activity- designed to enhance PA outcomes in children. | Tummy time- minutes/ day at baseline, post-intervention, and 6 months (GENEActiv accelerometer) | I > C for tummy time duration at post-intervention. | High |
| Moir et al., (2016) New Zealand [49] | RCT NR | N = 802, 0–2 years Yes | Maternity hospital, parent facilitated. 24 months. *Practical component*: family physical activity- active group sessions with infants at 3, 9, and 18 months of age. Child tummy time-infant prone to play/day. *Theory component*: educational sessions with parents focused on the importance of active play from birth and ideas for encouraging activity at different ages. Parents took home written resources to complement the sessions. | • Parent facilitated activities to enhance children's PA outcomes, measured post-intervention. • Structured activities with basic instructions for parents. | Tummy time (no of times & duration/day at 4 & 6 months. (Parent-reported questionnaires) Physical activity at 2 years/ toddlers (waist worn Actical accelerometers—5-days (24 hr) | I = C for tummy time (amount) at 4 and 6 months. I = C for time in light-to-vigorous activity at 2 years. | High |
| colspan Studies in primary school-aged children (6 to 12 years old) | | | | | | | |
| Brown et al., (2013) U.S. [44] | RCT SCT | N = 64, 10–14 years No | Primary/after school. 3 months. *Practical component*: traditional activities (horseback riding, dancing, hunting, hiking, camping and berry picking), storytelling and native games (double ball, ring the stick, running). *Theory component*: goal setting, sharing of success and challenges, goals achieved, discussing weekly activities (reflecting), and the importance of being active. | • Children played a central role in the activity. • The traditional activities promoted PA outcome. • The facilitator(s) asked questions to invoke creative thinking. | MVPA (avg min/d) and estimated energy expenditure (avg kcal/d) (Actical accelerometer). Self-reported measure: Physical activity/ behaviour (Modifiable PA questionnaire) | I > C for PA behaviour (MVPA and estimated energy expenditure). I > C PA behaviour | High |

(*Continued*)

**Table 3.** (*Continued*)

| Authors, (Year) Country | Study Design/ Theory used | Sample Size, Age/Grade, Parent Involvement | Intervention (setting, duration, experiential learning practical/theory components) | How experiential learning criteria were met | Measures/tools | Results | Overall Risk of Bias |
|---|---|---|---|---|---|---|---|
| Chen et al., (2009) U.S. [27] | RCT SCT | N = 67 8–10 years Yes | Family-based/after-school. 8 weeks. *Practical component*: 15 min/session of physical activities/ children's energy expenditure- dancing, brisk walking and jumping rope. 30 min/session of interactive activities (playing games) to promote/ motivate change in health behaviours. *Theory component*: educational/motivational workshops for enhancing children's knowledge and self-efficacy, self-regulation, critical thinking and problem-solving skills related to PA. | • The activities required children to physically complete the learning activity. • Activity designed to have a learning experience that promoted PA outcomes. • The facilitator(s) posed questions to invoke problem-solving or reflection | Physical Activity (Caltrac personal activity monitor) PA self-efficacy (Health behaviour questionnaire). PA knowledge (5- item questionnaire). | I > C for PA behaviour. I > C for knowledge about PA. I = C for self -efficacy (active lifestyles) | High |
| Eather et al., (2013a) Australia [28] | CRCT SCT | N = 48; mean age = 10.9 years ± 0.7) No | Primary schools, 8 weeks. *Practical component*: 60 min/ week PA—fun and vigorous games (e.g., enjoyable fitness activities, small-sided invasion games, skipping challenges). Daily break-time physical activity program (recess and lunch). *Theory component*: PE lessons to improve children's knowledge, skills (self-regulation, goal setting and self-monitoring) and self-efficacy to perform fitness activities. Positive reinforcement to participate in vigorous PA. Social support from teachers, fellow students and parents. | • The activity required children's input/ children had to physically complete the learning activity. • Children had autonomy in completing the activity that required them to be creative, problem-solve or be reflective. • Activity designed to have a learning experience that promoted PA outcomes. • Activity had a learning task or skill as the outcome | Flexibility (Fitness/ Activity gram). Physical activity (Pedometers), Attitudes toward physical fitness tests (purpose-designed questionnaire). | I > C for flexibility, muscular strength, and muscular endurance. I = C for PA levels. I = C for attitudes (fitness testing, physical fitness self-concept and self-esteem. | High |
| Eather et al., (2013b) Australia [45] | CRCT SCT | N = 226, mean age 10.7 years ± 0.6 years No | Primary schools, 8 weeks. *Practical component*: 60 min/ week of physical activities—fun and vigorous games (e.g., enjoyable fitness activities, small-sided invasion games, skipping challenges). Daily break-time activity program (recess and lunch). *Theory component*: PE lessons to improve knowledge, skills (self-regulation, self-monitoring), and self-efficacy to perform fitness activities. Weekly goal setting and reflection tasks based on the HRF components. | • Children played a central role in the activity, • The facilitator(s) posed questions to invoke problem-solving, creative thinking or reflection. • Activity had a learning task or skill as the outcome | CRF (20 m shuttle run test). Muscular fitness (Standing jump, 7-stage sit-up, basketball throw and push-up tests). Flexibility (sit and reach test) Physical activity- 7 days. (Yamax pedometers) | After 6-months, I > C for CRF (flexibility, muscular fitness and physical activity (steps/day) I = C for 3 measures of MF (basketball throw, push-ups and standing jump) | High |

(*Continued*)

**Table 3.** (Continued)

| Authors, (Year) Country | Study Design/ Theory used | Sample Size, Age/Grade, Parent Involvement | Intervention (setting, duration, experiential learning practical/theory components) | How experiential learning criteria were met | Measures/tools | Results | Overall Risk of Bias |
|---|---|---|---|---|---|---|---|
| Habib-Mourad et al., (2014) Lebanon [46] | RCT SCT | N = 387, 9–11 years Grades 4–5 No | Primary schools. 3 months. *Practical component*: 12 classroom physical activity sessions using fun and interactive activities to increase knowledge and skills. Playing games at recess. *Theory component*: physical education lessons, role modelling of significant others and positive reinforcements via praise and tokens. | • Children had played a central role in completing the activity that required them to be creative, problem-solve or be reflective. • Activity designed to have a learning experience that promoted PA outcomes. • Structured activities had learning task or skill as the outcome. | MVPA (Physical activity questionnaire) | I = C for MVPA (play at recess) | High |
| Lachausse, (2017) U.S. [50] | CRCT NR | N = 275 5–10 years Grades 4–6 No | Primary schools. 4 months. *Practical component*: aerobic exercises and outside play. *Theory component*: physical education (PE) lessons to promote physical activity. | • Children had autonomy in completing the activity that required them to be creative, problem-solve or be reflective. • Activity designed to have a learning experience that promoted PA outcomes. | Self-reported frequency of aerobic exercise and no. of days played outside in the past 7 days (Physical Activity Survey). | I = C for both aerobic exercise (60 min in 7 days) and the number of days played outside (in 7 days). | Some Concerns |
| Ourda et al., (2017) Greece [26] | CRCT NR | N = 112 6–12 years Grades 1–4 No | Primary schools. 2½ months. *Practical component*: 12 PA sessions, 35–40 mins. Balancing on two different body limbs, balancing a balloon, and 6 exercises lasting 7–8 minutes each for motor coordination/ creativity (one warm-up and one cool-down exercise). *Theory component*: physical education lesson for knowledge about healthy lifestyle. | • The activity required children's input/ children had to physically complete the learning activity. • The activity required children to be creative | Knowledge and attitude toward healthy lifestyle (Health Lifestyle Evaluation Instrument) | I > C for knowledge about a healthy lifestyle. I = C for attitudes towards a healthy lifestyle. | High |
| Reppa et al., (2015) Greece [47] | CRCT SCT | N = 112. Mean age 9.6 years Grade 4. No | Primary school. 2 months. *Practical component*: 14 lessons of creative gymnastic activities, creative movement and theatrical—movement games. *Theory components*: most activities were done in pairs, and mistakes were allowed which was useful for children to find more and better solutions. The teacher provided supportive feedback in all activities. | • Children played a central role in the activity, • The activity required children's input/ children had to physically complete the learning activity. • Structured activities with basic instructions, facilitated by a teacher. | Self-efficacy for physical activity. (PA Self-efficacy scale & pre-post questionnaire) | I > C for PA self-efficacy. | High |

*(Continued)*

**Table 3.** (Continued)

| Authors, (Year) Country | Study Design/ Theory used | Sample Size, Age/Grade, Parent Involvement | Intervention (setting, duration, experiential learning practical/theory components) | How experiential learning criteria were met | Measures/tools | Results | Overall Risk of Bias |
|---|---|---|---|---|---|---|---|
| Rosenkranz et al., (2010) U.S. [48] | CRCT SCT | N = 76 9–13 years. No | Scouts camp. 4 months. *Practical* component: 60–90 minutes of physically active recreation sessions (e.g., walking, dancing, yoga, and active games). *Theory component*: role modelling by peers and parents, skill building through active mastery experiences; enhancement of self-efficacy and proxy efficacy through role-playing and reinforcement of behaviour through verbal praise and merit badge. Leader encouragement of PA. | • Activity promoted PA outcomes. • The activity required children's input/ children had to physically complete the learning activity. • The facilitator(s) posed questions/ challenges to invoke problem-solving, creative thinking or reflection. | PA levels/minutes of MVPA. (Accelerometer). | I > C for moderate-to-vigorous physical activity) | Some Concerns |
| Siegrist et al., (2013) Germany [51] | CRCT NR | N = 724, Mean age 8.4 years Yes | Primary schools. 12 months. *Practical component*: 45 minutes session per month, 10 minutes of warm-up with running, playing games at high intensity, 30 minutes of exercises and 5 minutes of relaxation exercises. *Theory component*: education and encouragement on active and healthy lifestyles. *Other components*: Alteration of school environmental settings (e.g., the physical environment, organization of school breaks, playing during school time, and sports facilities) to promote more physical activity. | • Involved practical and reflection components- e.g., changes to the environment. • The activity required children's input to physically complete the learning activity. • Activities promoted PA outcomes. • The facilitator(s) posed questions to invoke problem-solving and creative thinking | Physical activity- no. of days of PA for 60 min/ day (MVPA index). Physical fitness-cardiopulmonary fitness, coordination, muscle strength, and flexibility (Munich fitness test). | I = C for PA (MVPA) I = C for physical fitness (cardiopulmonary fitness, coordination, muscle strength, and flexibility) | High |

CCM = chronic care model, CRCT = cluster randomised control trial, HRF = health-related fitness, I = intervention, C = control, LVA = light-to-vigorous activity, MVA = medium-to-vigorous activity, MVPA = moderate-to-vigorous physical activity, NR = not reported, Overall risk of bias = see S1 Table for Risk of Bias ratings on individual criteria, PA = physical activity, PE = physical education, RCT = randomised control trial, SCT = social cognitive theory.

settings, such as a scouts camp [48], maternity hospitals [49], primary care clinics [43], a community fitness centre [44], and family-based within homes [27]. Of the 12 studies, five studies [27,28,43,49,51] involved parents directly in the intervention activities with children while two studies [46,48] involved parents indirectly via awareness and information sessions. Both studies with children aged under five years employed parent-facilitated activities.

Most studies (10/12) had a high risk of bias [26–28,43–47,49,51], while two studies were graded as 'some concerns' [48,50]. The assessment domains with overall scores for each of the studies are presented in S1 Table.

## Experiential learning approaches

Practical components. The practical components of the experiential learning interventions were divided into five categories, which included: (1) outdoor and adventure activities (n = 1) [44]; (2) fitness activities and challenges (n = 7) [27,28,44,45,48,50,51]; (3) rhythmic and expressive movement (creative and theatrical) activities (n = 6)[27,44,47,48,50,51]; (4) modified sports and games (n = 6) [28,45,45,48,50,51]; and (5) active play (n = 3) [43,46,49].

**Theory components.** The theory components of the experiential learning interventions were divided into two categories: (1) behaviour change/motivational strategies, [52] and (2) information and education strategies. The majority of the studies (n = 8) used approaches from both categories [27,28,43–6,49,51], while two studies [47,48] used only behaviour change/motivational strategies and two studies [26,50] used only information/education strategies. One of the studies [51] used these two categories together with additional environmental/organisational change strategies (e.g., altered physical environment, organization of school breaks, playing during school time, and sports facilities/playground) to encourage more physical activity in children. Both the practical and theory components of the experiential learning intervention were considered in interpreting the intervention effects.

**Intervention effects.** The effect sizes for the experiential learning-based physical activity interventions on knowledge, attitudes and behavioural outcomes are presented in Table 4. The results of the heterogeneity analysis showed that there was moderate to high heterogeneity of effects across studies for physical activity behaviour outcomes. For physical activity behaviour outcome, the evidence of heterogeneity in effects was high (I2 = 79.9%) and (95% CI, -0.068 to -0.321), while for physical activity attitude outcome the heterogeneity in effects was low (I2 = 33.4%) with (95% CI, -0.003 to -0.549).

Furthermore, when the necessary analyses were performed in SPSS, the Q-statistics values (Q = 36.552, df = 9, p < 0.001) were found to be statistically significant (S2 Fig). In addition, Tau-squared, H-squared, and I-squared values were found to be 0.065, 4.985, and 79.9, respectively, showing statistically significant heterogeneity between studies for behaviour outcomes (see S2 Fig).

Low heterogeneity of effects was found across studies for attitude outcomes as shown by the Q-statistics (Q = 2.970, df = 2, p < 0.226) (see S3 Fig), with no statistical significance. For behaviour outcomes, visual inspection indicates that the included studies were evenly clustered within both sides of the funnel plot, with no evidence of publication bias, (Egger's test: t = -0.85, P < 0.4) (see S2 Fig). There are not enough valid cases for processing for Egger's test for attitude outcome, so no statistics were computed (see S3 Fig).

The trim and fill method suggested that no adjustments were required to correct the asymmetry of the funnel plot. Therefore, any potential publication bias was considered minimal. For physical activity attitude outcomes, there were not enough valid cases for processing thus no statistics were computed.

**Physical activity behavioural outcomes (children aged 1–5 years).** In children aged 1 to 5 years, two studies [43,49] measured intervention effects on physical activity behaviour change. Both interventions included infants together with their mothers/caregivers and used similar practical and theory experiential learning activities. For example, both studies included practical components of active play/tummy time and theoretical components targeted at parents including behavioural change/motivational strategies and information/education strategies. Although the experiential learning intervention components were similar, the study by Hewitt and colleagues [43] (d = 0.5) was informed by Social Cognitive Theory (SCT) and was only implemented for a short duration (one month) whereas the study by Moir and colleagues [49] (d = 0.2) implemented a 2-year intervention and did not report a theoretical

**Table 4. Experiential learning approaches and effect sizes on physical activity-related knowledge, attitudes, and behavioural outcomes.**

| Study | Experiential Learning activities (categories of practical and theory components) | | Mean effect size (Cohen's d) |
|---|---|---|---|
| | Practical | Theory | |
| **Preschool-aged children (< 5 years old)** | | | |
| *Outcome*: Behaviour | | | |
| Hewitt et al., 2020 [43] | Active play/ Tummy time | Behaviour change/motivational strategies Information/education strategies | Tummy time (duration) 0.5 |
| Moir et al. (2016) [49] | Active play/ Tummy time | Behaviour change/motivational strategies Information/education strategies | Tummy time (infant prone to play/day) 0.2 |
| **Primary school-aged children (6 to 12 years old)** | | | |
| *Outcome*: Behaviour | | | |
| Eather et al. (2013) a [28] | Fitness activities and challenges Modified sports and games. | Behaviour change/motivational strategies Information/education strategies | Flexibility levels 0.9 Muscular fitness 0.9 Physical activity levels 0 |
| Brown et al., (2013) [44] | Outdoor and adventure activities Fitness activities and challenges Rhythmic and expressive movement (creative and theatrical) activities Modified sports and games | Behaviour change/motivational strategies Information/education strategies | MVPA, Energy expenditure (avg kcal/d) 0.8 |
| Chen et al. (2009) [27] | Fitness activities and challenges Rhythmic and expressive movement (creative and theatrical) activities | Behaviour change/motivational strategies Information/education strategies | Physical activity (behaviour) 0.6 |
| Rosenkranz et al. (2010) [48] | Fitness activities and challenges Rhythmic and expressive movement (creative and theatrical) activities Modified sports and games | Behaviour change/motivational strategies | Days/week of 60 min MVPA 0.5 |
| Eather et al. (2013) b [45] | Fitness activities and challenges Modified sports and games | Behaviour change/motivational strategies Information/education strategies | Health-related fitness and physical activity levels 0.4 |
| Siegrist et al. (2013) [51] | Fitness activities and challenges Rhythmic and expressive movement (creative and theatrical) activities Modified sports and games | Behaviour change/motivational strategies Information strategies | Physical activity (MVPA) 0.3 Physical fitness 0.3 |
| Lachausse, (2017) [50] | Fitness activities and challenges Modified sports and games | Information/education strategies | Aerobic exercise 0.1 Outdoor play 0.1 |
| Habib-Mourad et al. (2014) [17] | Active play | Behaviour change/motivational strategies Information/education strategies | Physical activity -recess and after school 0.04 |
| *Outcome*: Attitude | | | |
| Chen et al. (2009) [27] | Fitness activities and challenges Rhythmic and expressive movement (creative and theatrical) activities | Behaviour change/motivational strategies Information/education strategies | Physical activity self-efficacy 0.5 |
| Reppa et al. (2015) [47] | Rhythmic and expressive movement (creative and theatrical) activities. | Behaviour change/motivational strategies | Physical activity self-efficacy 0.4 |
| Ourda et al. (2017) [26] | Rhythmic and expressive movement (creative and theatrical) activities. | Information/education strategies | Physical activity attitudes 0.01 |
| Eather et al. (2013) a [28] | Fitness activities and challenges Modified sports and games | Behaviour change/motivational strategies Information/education strategies | Insufficient reported data* |
| *Outcome*: Knowledge | | | |

*(Continued)*

**Table 4.** (Continued)

| Study | Experiential Learning activities (categories of practical and theory components) | | Mean effect size (Cohen's d) |
|---|---|---|---|
| Ourda et al. (2017) [26] | Rhythmic and expressive movement (creative and theatrical) activities. | Information/education strategies | Physical activity knowledge 1.3 |
| Chen et al. (2009) [27] | Fitness activities and challenges Rhythmic and expressive movement (creative and theatrical) activities | Behaviour change/motivational strategies Information/education strategies | Physical activity knowledge 0.4 |

MVPA; Moderate to vigorous physical activity, (*Effect sizes could not be calculated due to missing data). The studies are ordered from largest to smallest effect.

framework. No studies among young children measured children's attitudes towards and knowledge of physical activity.

**Physical activity behavioural outcomes (children aged 6–12 years).** In primary school-aged children (6–12 years), eight studies assessed the effects of the intervention on physical activity behaviour change [27,28,44–46,48,50,51]. Of these eight studies, two [28,44] reported large effects, four [27,45,48,51] reported medium effects and two [46,50] reported small effects on children's physical activity behavioural outcomes.

Of the two studies with large effects, Eather et al. [28] reported the largest effect ($d$ = 0.9) on children's fitness outcomes following their school-based fitness and health education intervention. Brown et al. [44] reported an effect size of $d$ = 0.8 on children's physical activity (MVPA) and energy expenditure (avg kcal/d), in their intervention conducted after school in classrooms and community and fitness centres. Both interventions included fitness activities and challenges and modified sports and games as part of their practical experiential learning activities. However, Brown et al. [44] also included outdoor and adventure activities and rhythmic and expressive movement activities. Both studies had an intervention duration of 2–3 months, were based on SCT and one study [28] involved parents (e.g., as social support) in the intervention activities. Additionally, the theory component of the intervention involved behavioural change/motivational strategies (goal setting, reflecting, self-regulation, positive reinforcement) and information/education strategies for both studies.

The four studies that reported medium effects on children's physical activity behavioural outcomes [27,45,48,51] (M$d$ = 0.5), all included fitness activities and challenges. In addition, three of the four studies used modified sports and games and three used rhythmic and expressive movement activities. All these studies included both theory components in the intervention (behavioural change/motivational and information/education strategies), except one study [48], that used only behavioural change/motivational strategies. Three of the four studies had an intervention duration ranging from 2–3 months and were based on SCT. Two of these studies did not involve parents directly in the intervention activities [45,48] while Chen et al. [27] enrolled parents as partners in behavioural change in their children. In contrast, Siegrist et al. [51] ($d$ = 0.3) involved a 12-month intervention and involved parents (e.g., training and motivations to be more active with their children) in the intervention. No theoretical framework was mentioned for this study.

The remaining two studies that measured behavioural outcomes [46,50], reported a small mean effect (M$d$ = 0.1). The practical components of Lachausse et al.'s [50] intervention included fitness activities and challenges and modified sports and games, but the theoretical component only included information/educational approaches. In contrast, Habib-Mourad et al.'s [46] intervention only included active play activities. Both behaviour change and information/educational theoretical approaches were used to enhance active play activities.

**Physical Activity Attitudes (children aged 6–12 years).** Four studies conducted with primary school-aged children measured changes in physical activity-related attitudes [26–28,47].

Effect sizes were able to be calculated for three studies (the effect size could not be calculated for the fourth study due to missing data), of which two studies [27,47] reported a medium mean effect (M$d$ = 0.4) and one study [26] had small an effect ($d$ = 0.01). All three interventions [26,27,47] included rhythmic and expressive movement practical activities; one of the interventions also included fitness activities and challenges [27]. Two out of three studies [27,47] included behaviour change/motivational strategies in the intervention, one of which also included information/education strategies [27]. In contrast, one study only included information/education strategies in the theory component of the intervention [26]. All three interventions were delivered for a similar duration (2 months). Two interventions were based on SCT, whereas one intervention [26] ($d$ = 0.4) did not report a theoretical base. Likewise, two interventions did not involve parents, whereas one did [27] ($d$ = 0.3).

**Physical activity knowledge (children aged 6–12 years).** Two studies [26,27] measured change in primary school-aged children's knowledge regarding physical activity. One of the studies [26] reported a large effect ($d$ = 1.3) on increasing knowledge about an active lifestyle. This intervention included only information/education-focused theory strategies and rhythmic and expressive movement activities. The other study that assessed change in primary school-aged children's physical activity knowledge reported a medium effect [27] (M$d$ = 0.4). This intervention was based on SCT, used a combination of information/education and behaviour change theory strategies, and used two or more practical components. Both studies had an intervention duration of 2–3 months. While the intervention by Chen et al. [27] included parents, the other interventions did not.

## Discussion

This review aimed to summarise the effectiveness of physical activity-related experiential learning interventions conducted in pre-schools, primary schools and community settings, on physical activity outcomes (i.e., behaviour, knowledge and attitudes) in children aged birth to 12 years. Most of the included intervention studies were conducted in primary school-aged children (n = 10). Of the studies among school-aged children for which effect sizes could be calculated, six out of eight displayed medium to large effects for improving physical activity behaviour, and two out of three studies displayed medium effects for improving physical activity-related attitudes. Likewise, the two studies that assessed effects on physical activity-related knowledge outcomes among school-aged children displayed medium to large effects. Additionally, the two studies that used experiential learning interventions involving parents and their infants demonstrated small to medium effects for promoting infants' physical activity behaviour (i.e., tummy time).

A number of factors may have contributed to the positive changes in physical activity behaviours in the studies among school-aged children. The type of intervention components may have been influential. In this review, all of the experiential learning activities comprised a practical component and a theory component, however in the studies that had more favourable outcomes (i.e., larger effect sizes) two or three practical components (e.g., fitness activities and challenges, and modified sports and games, and rhythmic and expressive movements) and at least two theory components (e.g., behaviours change/motivational strategies and information/education strategies) were used. The combination of a number of different types of activities complemented with a number of different behaviour change techniques may have been important in the success of these interventions. The combination of activities may have appealed to a more diverse group of children, hence potentially contributing to greater changes in the overall intervention outcomes. It is well documented that primary school-aged children have a number of different learning styles and it is important for teachers and educators to

differentiate to ensure that all students have the opportunity to learn and achieve [53,54]. The multiple practical and theory components implemented in the more successful studies in this review may have enabled broader learning and engagement from all students involved in the intervention.

The type of activities may also have been a factor. A number of the most effective interventions included in this review used modified games and challenges. For example, Eather et al [28] used fun and enjoyable games that comprised small-sided games and skipping challenges and Chen et al. [27] and Brown et al. [44] included multiple practical activities such as dancing, brisk walking, running, and jump rope. In relation to the theory components, effective studies used various behaviour change/motivational strategies to influence children. For example, Eather et al [28] enhanced children's self-regulation, goal-setting and self-monitoring skills to improve children's knowledge of and self-efficacy towards physical activity as well as positive reinforcements to enable children to participate in physical activity. On the other hand, Brown et al. [44] used techniques such as goal-setting, and reflection (sharing of success, challenges, and goals achieved) to support children in being physically active. The combination of practical and theory components aligns closely with key attributes of experiential learning activities, as described above [28,44]. For the most successful studies, children played a central role in the activities and were encouraged to explore and engage in the activities offered. Furthermore, the activities required children's input and children had to physically do something as part of the learning activity and were encouraged to problem-solve or be reflective. Although including several practical and theory components in an intervention is more time-consuming and perhaps more expensive, the combination of components may be pivotal in modifying children's physical activity behaviour. Experiential learning activities have also been shown to be effective in changing other behaviours in primary-aged children, including healthy eating behaviours [30] further supporting the importance of a combination of practical and theory components in interventions.

Although a combination of intervention components (i.e., experiential learning) would seem important for changing physical activity behaviour in children, it should be noted that some studies that included a single component (a practical or theory component) have also been successful in changing behaviour. For example, a review of non-experiential learning school-based studies [55] suggested that the provision of fun and enjoyable activities contributed to the change in children's physical activity behaviour. The interventions typically did not involve a theory component. Other aspects, such as the intensity of the interventions may have also contributed to the change in children's physical activity behaviour in the studies reviewed. Most effective interventions in this review were implemented intensely over a relatively short duration of time (i.e., less than four months) and were underpinned by an interpersonal behaviour change theory such as Social Cognitive Theory. These components have also been highlighted as key characteristics of interventions in other non-experiential physical activity interventions targeting primary school-aged children. For example, reviews by Ahmed et al. (2021) and Nixon et al. (2012) specifically mentioned that studies that were more intense and delivered over a relatively short period were more effective in changing children's physical activity behaviour.

Furthermore, the changes in physical activity behaviour may have been an indirect result of changes in behavioural constructs such as self-efficacy. For example, Chen et al.'s [27] intervention improved children's self-efficacy (an attitudinal outcome), which may have subsequently improved their physical activity behaviour. This is consistent with Social Cognitive Theory, in that enhancing self-efficacy may help influence behaviour change [56]. Self-efficacy is defined as the belief in one's abilities to successfully engage in a particular behaviour [57]. It is however difficult to disentangle the necessary components of an effective experiential

learning intervention, but the inclusion of multiple activities, a combination of practical and theory components, and the use of effective behaviour change theory to underpin intervention development are potentially important.

Only four studies in this review evaluated the effects of experiential learning interventions on children's physical activity-related attitudes. Two of these studies also assessed the effects on children's knowledge. In relation to attitudinal outcomes, two of the effective studies were successful in improving children's physical activity self-efficacy [27,47]. These studies used several strategies in the intervention to enhance self-efficacy such as improving children's self-regulation capabilities related to physical activity (e.g., critical thinking and problem-solving skills) [27] and social support (e.g., small groups/pairs and positive feedback and encouragement) [47]. According to other researchers, the characteristics identified in these intervention studies, such as self-regulation [58,59] and social support [60,61] could enhance physical activity self-efficacy in children. Self-efficacy can play a major role in children's motivation to participate in physical activity [62]. Studies show that children who have higher self-efficacy are more confident in choosing to participate in physical activity compared to children with low self-efficacy [63,64]. A review of the determinants of change in physical activity in children found consistent positive associations between self-efficacy and physical activity in children [65].

It is not surprising that the intervention that had the largest effect on children's physical activity-related knowledge focused exclusively on information/education strategies in the intervention [26]. This study focused on the development of children's motor creativity through physical education lessons, and playful motor creative activities using physical activity stimuli to improve knowledge about physical activity. This improvement in knowledge could be due to the intervention being specifically designed to assess the effect of motor creativity in promoting children's physical activity knowledge.

Future experiential learning interventions aiming to improve physical activity-related attitudes and knowledge amongst children may look to apply self-efficacy-enhancing strategies such as improving children's self-regulation capabilities, and social support and enhancing children's motor creativity through playful activities. Physical activity knowledge and attitudes are important outcomes because they may determine children's beliefs in their ability (i.e., their perceived self-efficacy) to participate, and subsequently, their effort and persistence to be physically active [66].

It is difficult to draw conclusions from the studies that targeted preschool-aged children as the number of studies was small (i.e., only two studies). As expected, both studies involved parents and both studies comprised an active play/tummy time practical component and utilised behavioural change/motivational strategies and information/education strategies. Although definitive conclusions cannot be drawn from these studies, what is known is that physical activity should be encouraged from birth and that physical activity should be part of everyday routines for young children independent of the time that they spend at home or in formal care. To support this notion, several countries [6–8], as well as the World Health organization, have released 24-hour movement guidelines for young children which describe the type, intensity and duration of physical activity for young children, highlighting the importance of tummy time for infants. Furthermore, for younger children, the importance of including both parents and children in experiential learning interventions may be even more important than for older children as the child's physical activity may be influenced by parents' knowledge, attitudes and behaviours [67,68].

The results of this study should be considered in light of some of the challenges in conducting such a review, including the following: 1) the experiential learning activities (practical and theory components) were combined into categories to assist in grouping the different

intervention approaches, however this may not have completely reflected the interventions; 2) there was considerable diversity in the specific activities and theory-related components (e.g., some interventions included structured sessions while others were less structured, some interventions included goal setting within their behaviour change activities while others did not); 3) limited information was reported about the fidelity or delivery of the interventions and the engagement of participants, and; iv) most studies had a high risk of bias suggesting that they had methodological flaws. However, despite these challenges some recommendations can be made from this review that may improve children's physical activity-related knowledge, attitudes and behaviours through experiential learning interventions, particularly if enjoyable and engaging practical activities are combined with effective educational approaches and evidence-based behaviour change techniques.

## Implications for research and interventions

The findings of this review have important implications for future experiential learning interventions, indicating that it may be important to consider: 1) combining physical activity practical components with theory components such as behaviour change/educational strategies; 2) incorporating multiple practical components and multiple theory components; 3) designing interventions based on relevant behaviour change theories; and 4) having a high intensity intervention (more number of sessions/activities). Additionally, more studies are needed that explore the effect of experiential learning activities on physical activity knowledge and attitudes in school-aged children. More studies are also needed that investigate the effects on physical activity knowledge, attitudes and behaviours in preschool-aged children. Also, further investigation exploring the specific components of practical and theory approaches that are successful in changing behaviour, knowledge and attitudes is warranted. Future interventions should consider clearly specifying how experiential or practical components are incorporated into the intervention, and the level of children's participation in intervention activities. Further investigation into the longer-term impacts of experiential learning activities and maintenance of the behaviour change is required.

## Strengths and limitations

The present review included studies with a broader age group and experiential learning interventions delivered outside of school settings. We used broad search terms and a comprehensive inclusion criterion, which yielded many eligible studies that were independently screened by two reviewers. Only RCT and CRCT studies with experiential learning interventions were included, which enhances the internal validity of the review. We calculated effect sizes (Cohen's d) to quantify the relative effect of the intervention strategies on the outcomes across age groups, which has not been done previously. We also assessed the risk of bias using the Cochrane Collaboration tool, which was important for highlighting methodological gaps in the evidence base.

There were some limitations associated with this review. This review only included papers published in English and, therefore did not include papers published in other languages. Evidence from non-RCTs was excluded but may still provide useful insights about the effectiveness of experiential learning interventions. The risk of bias assessments of the studies was generally high, therefore the strength of the conclusions from this review may need to be considered carefully. There was evidence of heterogeneity in the effects of physical activity behaviour outcomes and, to a lesser extent, on attitude outcomes. This is likely influenced by differences in study outcomes (e.g., physical activity behaviour outcomes included MVPA, aerobic exercise, steps per day, and tummy time), interventions, and other methodological

differences (e.g., different lengths of assessment follow-up). Additionally, moderate to high statistical heterogeneity was observed for physical activity behaviour outcomes which should be considered when interpreting findings. Lastly, this review is limited to the effects of experiential learning activities on physical activity outcomes only, and therefore findings are not generalised to other lifestyle behaviours.

## Conclusion

This review provides some evidence supporting the effectiveness of experiential learning interventions in improving physical activity outcomes in school-aged children. From studies for which effect sizes could be calculated, six out of eight showed medium to large effects for behavioural outcomes, all three studies that assessed attitudinal outcomes displayed medium effects, and both studies that assessed knowledge outcomes displayed medium to large effects. Additional evidence is needed in children under five years, but the two parent-facilitated interventions involving infants demonstrated small to medium effects on behaviour change. Effective interventions typically combined enjoyable practical activities (e.g., fitness activities, games and challenges), with behaviour change techniques (e.g., goal setting, self-monitoring and reflection), were underpinned by a behaviour change theory, and were often short in duration (e.g., < 4 months) but intense (e.g., several sessions/week). Future experiential learning interventions need to strengthen the evidence with rigorous methodological quality and clear reporting of the experiential learning components (practical and theory) as well as assess the maintenance of the changes over time.

## Supporting information

**S1 Fig. PRISMA flow chart for study selection.**
(TIF)

**S2 Fig. Meta-analysis output for behaviour outcome.**
(DOCX)

**S3 Fig. Meta-analysis output for attitude outcome.**
(DOCX)

**S1 Table. Quality assessment of the included studies.**
(TIF)

**S2 Table. PRISMA 2020 checklist.**
(DOCX)

## Acknowledgments

We would like to thank Grace Norton (GN) for her help with study selection and Zhiguang Zhang for her assistance in quality appraisal.

## Author Contributions

**Conceptualization:** Rachel A. Jones, Bridget Kelly, Dylan P. Cliff.

**Data curation:** Sumantla D. Varman, Rachel A. Jones, Bridget Kelly, Dylan P. Cliff.

**Formal analysis:** Sumantla D. Varman, Rachel A. Jones.

**Methodology:** Sumantla D. Varman, Rachel A. Jones, Bridget Kelly, Dylan P. Cliff.

**Project administration:** Sumantla D. Varman.

**Supervision:** Rachel A. Jones, Bridget Kelly, Dylan P. Cliff.

**Validation:** Rachel A. Jones.

**Writing – original draft:** Sumantla D. Varman, Dylan P. Cliff.

**Writing – review & editing:** Sumantla D. Varman, Rachel A. Jones, Bridget Kelly, Megan L. Hammersley, Anne-Maree Parrish, Rebecca Stanley.

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
