## [Decision Letter · Decision Letter 0]

25 Jul 2023

PONE-D-23-02390The effect of experiential learning interventions on physical activity outcomes in children: A systematic reviewPLOS ONE

Dear Dr. Varman,

Thank you for submitting your manuscript to PLOS ONE. After careful consideration, we feel that it has merit but does not fully meet PLOS ONE’s publication criteria as it currently stands. Therefore, we invite you to submit a revised version of the manuscript that addresses the points raised during the review process.

Your manuscript has been evaluated by two reviewers, and their comments are appended below. Both reviewers have requested clarifications regarding the study design and methodology; specifically the categories used, the determination of the studies as 'experiential learning interventions', and the possibility of quantitative meta-analytic analysis. Please ensure you address each of the reviewers' comments when revising your manuscript.

We look forward to receiving your revised manuscript.

Kind regards,

Hugh Cowley

Staff Editor

PLOS ONE

Reviewers' comments:

Reviewer's Responses to Questions

**Comments to the Author**

1. Is the manuscript technically sound, and do the data support the conclusions?

Reviewer #1: Yes

Reviewer #2: Partly

2. Has the statistical analysis been performed appropriately and rigorously? 

Reviewer #1: Yes

Reviewer #2: No

3. Have the authors made all data underlying the findings in their manuscript fully available?

Reviewer #1: Yes

Reviewer #2: Yes

4. Is the manuscript presented in an intelligible fashion and written in standard English?

Reviewer #1: Yes

Reviewer #2: Yes

5. Review Comments to the Author

Reviewer #1: Wonderful job on the writing and analysis. A few points to consider:

1) Under the Interventions heading (line 120)"...The activity went beyond the provision of information...." What does that mean exactly? If this is referencing the intervention required a component such as reflection, please be specific as that line is very vague.

2) When comparing practical vs theory- the practical category provides a very detailed description of what it includes (because there are 5 categories). However the categories behavior change/motivational strategies and information/education are very broad (and only 2 categories). Is there a way to separate those categories in more detail? Perhaps in Table 2 make behavior change a category (and provide examples); motivational strategies a category (and provide examples); etc. These broad categories make Table 4 vague on the theory component. The majority of the studies have both theory components- there needs to be information of what aspects of the theory were employed.

Reviewer #2: Thank you for asking me to review this manuscript. I was very interested by the research topic but there are a few fundamental flaws in the research design and reporting that will need to be addressed before I'd be willing to support its publication in PLOS ONE.

The first, and most pressing issue, is that I can not derive how the interventions captured by the systematic review were determined to meet the fidelity requirement of an 'experiential learning intervention'. It appears that the only filter was the Search Criteria. The Quality Appraisal process does not seem to address the fidelity and external validity issue either. I would ask that in Table 3, the authors consider including a justification for each intervention as to how the fidelity of being a 'Experiential Learning' intervention were met.

Secondly, if the authors wish to report beyond the qualitative aspects of these interventions with the effect sizes and insight that draws, then they should be provide a far more detailed meta-analytic analysis. This is well within capabilities of these authors but at the very least they should include analyses that included tests of heterogeneity and statistical publication bias.

6. PLOS authors have the option to publish the peer review history of their article (what does this mean?). If published, this will include your full peer review and any attached files.

Reviewer #1: **Yes: **April Ruther

Reviewer #2: No

---

## [Author Response · Author response to Decision Letter 0]

22 Sep 2023

Reviewer 1 comment: 

1) Under the Interventions heading (line 120)"...The activity went beyond the provision of information...." What does that mean exactly? If this is referencing the intervention required a component such as reflection, please be specific as that line is very vague.

Response:

Yes, this is referring to the interventions requiring reflection of the activity e.g., through discussions. The following has been added to the text under the interventions heading. i.e., it involved practical components and activities with a focus on a learning task

Reviewer comment:

2) When comparing practical vs theory- the practical category provides a very detailed description of what it includes (because there are 5 categories). However, the categories behavior change/motivational strategies and information/education are very broad (and only 2 categories). Is there a way to separate those categories in more detail? Perhaps in Table 2 make behavior change a category (and provide examples); motivational strategies a category (and provide examples); etc. These broad categories make Table 4 vague on the theory component. The majority of the studies have both theory components- there needs to be information of what aspects of the theory were employed.

Response:

In Table 2, We have separated these and made behaviour change a category (and provided examples); and motivational strategies a category (and provided examples); and added one more category such as reflection with examples. We have also separated information/education and provided examples to add more detail.

Reviewer 2 comment: 

1) The first, and most pressing issue, is that I can not derive how the interventions captured by the systematic review were determined to meet the fidelity requirement of an 'experiential learning intervention'. It appears that the only filter was the Search Criteria. The Quality Appraisal process does not seem to address the fidelity and external validity issue either. I would ask that in Table 3, the authors consider including a justification for each intervention as to how the fidelity of being a 'Experiential Learning' intervention were met.

Response:

The criteria for an experiential learning intervention were described in the methods as follows:

Studies were included where the intervention incorporated an experiential learning activity, with one or more of the following characteristics (1) children played a central role in the activity, allowing them to engage with, and explore, the phenomena; (2) the activity went beyond the provision of information, instruction, encouragement, equipment or change to the environment, i.e. it involved and practical and reflection components; (3) the activity required children’s input and children had to physically do something as part of the learning activity; (4) the children had a level of autonomy in completing some part of the activity that required them to be creative, problem-solve or be reflective; (5) the activity was specifically designed to have a learning experience that enhanced physical activity outcomes, measured post-intervention; (6) the activities had a clear learning task or skill as the outcome; (7) the activity was coordinated/facilitated by a leader such as a teacher or educator, or parent; and (8) the facilitator(s) provided the structure for the activity such as basic instruction, posing questions to invoke problem-solving, creative thinking or reflection [31-33]

A column has been added in Table 3 indicating how each intervention met the key experiential learning criteria.

Reviewer 2 comment: 

2) Secondly, if the authors wish to report beyond the qualitative aspects of these interventions with the effect sizes and insight that draws, then they should be provide a far more detailed meta-analytic analysis. This is well within capabilities of these authors but at the very least they should include analyses that included tests of heterogeneity and statistical publication bias.

Response:

Additional analyses have been conducted to include tests of heterogeneity and statistical publication bias. 

The following text has been added to the Manuscript. 

Data analysis:

Where there were at least three included studies for a group of outcomes (behaviour, knowledge, or attitudes), heterogeneity was determined by Cochran’s I2 Statistic. I2 values of 25, 50, and 75 were considered to indicate low, moderate and high heterogeneity, respectively (Deeks, et al., 2005). Publication bias was determined by Egger's regression test and visual inspection of funnel plots (Higgins, et al., 2003).

Results:

For physical activity behaviour outcomes, there was no evidence of publication bias (p = 0.42), but there was evidence of high heterogeneity in effects (I2 = 0.84). For physical activity attitude outcomes, there was no evidence of publication bias (p = 0.88) and the heterogeneity in effects was low to medium (I2 = 0.33).

Discussion:

Limitations- There was evidence of heterogeneity in the effects for physical activity behaviour outcomes and, to a lesser extent, for attitude outcomes. This is likely influenced by differences in study outcomes (E.g., physical activity behaviour outcomes included MVPA, aerobic exercise, steps per day, and tummy time), interventions, and other methodological differences (e.g., different lengths of assessment follow-up).

---

## [Editor Report · Decision Letter 1]

27 Sep 2023

PONE-D-23-02390R1The effect of experiential learning interventions on physical activity outcomes in children: A systematic reviewPLOS ONE

Dear Ms Varman,

Thank you for submitting your manuscript to PLOS ONE. After careful consideration, we feel that it has merit but does not fully meet PLOS ONE’s publication criteria as it currently stands. Therefore, we invite you to submit a revised version of the manuscript that addresses the points raised during the review process.

We look forward to receiving your revised manuscript.

Kind regards,

Dean Dudley, PhD

Guest Editor

PLOS ONE

Journal Requirements:

Additional Editor Comments (if provided):

I want to thank the authors for their revisions to this manuscript. It is already a much stronger study now. That said, I still have a number of concerns with the limited measures and interpretation of the heterogeneity and publication bias results. First, I2 alone is not the most transparent way of reporting heterogeneity. I would expect that as a minimum, that the Q, Tau, and prediction interval statistics are reported so that the reader can determine more than simply the proportion of observed variance attributed to actual changes in effect sizes rather than sampling error. The Q statistic tests the null hypothesis of whether all studies in the model share a common effect size. The T2 statistic determines the variance of true effect sizes, while the prediction interval provides a confidence interval (with 95% confidence limits) that encompasses the range of true effect sizes for all observed samples.

Finally, the Egger's regression test is only one means of reporting publications bias and my recommendation is that at least measures are used. My suggestion would be to also include the Classic Fail Safe N or the Trim and Fill method.

---

## [Author Response · Author response to Decision Letter 1]

11 Nov 2023

Journal Requirements

Please review your reference list to ensure that it is complete and correct. If you have cited papers that have been retracted, please include the rationale for doing so in the manuscript text or remove these references and replace them with relevant current references. Any changes to the reference list should be mentioned in the rebuttal letter that accompanies your revised manuscript. If you need to cite a retracted article, indicate the article’s retracted status in the References list and also include a citation and full reference for the retraction notice.

Response

The reference has been reviewed thoroughly and the DOI has been added to most of the references. 

Additional editor comments

I want to thank the authors for their revisions to this manuscript. It is already a much stronger study now. That said, I still have a number of concerns with the limited measures and interpretation of the heterogeneity and publication bias results. 

First, I2 alone is not the most transparent way of reporting heterogeneity. I would expect that as a minimum, that the Q, Tau, and prediction interval statistics are reported so that the reader can determine more than simply the proportion of observed variance attributed to actual changes in effect sizes rather than sampling error. The Q statistic tests the null hypothesis of whether all studies in the model share a common effect size. The T2 statistic determines the variance of true effect sizes, while the prediction interval provides a confidence interval (with 95% confidence limits) that encompasses the range of true effect sizes for all observed samples.

Finally, Egger's regression test is only one means of reporting publications bias and my recommendation is that at least measures are used. My suggestion would be to also include the Classic Fail-Safe N or the Trim and Fill method.

Response

Additional analyses have been conducted for heterogeneity and publication bias. 

The following text has been added to the Manuscript regarding the additional tests and interpretation of the results.

Abstract- Results- Moderate to high statistical heterogeneity was observed for behaviour outcomes.

Methods

Data analysis:

The text on heterogeneity has been amended/replaced.

Possible causes of heterogeneity in effects were assessed using the I2 statistic, Cochran Q-tests and subgroup analyses. Cochran Q-tests were used to determine the statistical heterogeneity for physical activity behaviour and attitude outcomes, with statistical significance set at p < 0.1. All subgroup analyses used a single pooled estimate of Tau-squared (between-study variance). Random effects models were employed to allow for variation between studies. The confidence interval (with 95% confidence limits) was used to show the range of true effect sizes for included studies.

The text on publication bias has been amended/replaced.

Potential publication bias was assessed by inspecting forest and funnel plots as well as via running Egger’s Test, [39], (using a recommended criterion of p < 0.1, [40] Additionally, Duval and Tweedie’s Trim and Fill method was employed to estimate and impute potential unpublished effects to assess their impact on the pooled effect size and significance [41]. Subgroup analyses could not be performed for knowledge outcomes due to the small number of studies. Additionally, the trim and fill test for physical activity attitude outcome was not run due to not enough valid cases for processing thus no statistics were computed. The calculations for heterogeneity tests, subgroup analysis and publication bias evaluation were performed using the IBM SPSS Statistics 28.0 and the guidance from (Sen and Yildirim, 2022).

Results:

Intervention effects-

The text in the results section has been amended/replaced. We have additionally reported the results of the Tau-squared and Cochran Q tests to show heterogeneity.

The results of the heterogeneity analysis showed that there was moderate to high heterogeneity of effects across studies for physical activity behaviour outcomes. For physical activity behaviour outcome, the evidence of heterogeneity in effects was high (I2 = 79.9%) and (95% CI, -0.068 to -0.321), while for physical activity attitude outcome the heterogeneity in effects was low (I2 = 33.4%) with (95% CI, -0.003 to -0.549).

Furthermore, when the necessary analyses were performed in SPSS, the Q-statistics values (Q =36.552, df = 9, p < 0.001) were found to be statistically significant (see Fig 1 in S1 Fig).

In addition, Tau-squared, H-squared, and I-squared values were found to be 0.065, 4.985, and 79.9, respectively, showing statistically significant heterogeneity between studies for behaviour outcomes (see Fig 2 in S1 Fig). 

Low heterogeneity of effects was found across no studies for attitude outcomes as shown by the Q-statistics (Q =2.970, df = 2, p < 0.226) (see Fig 2 in S1 Fig), with no statistical significance.

For behaviour outcomes, visual inspection indicates that the included studies were evenly clustered within both sides of the funnel plot, with no evidence of publication bias, (Egger’s test: t = -0.85, P < 0.4) (see Fig 3 in S1 Fig). There are not enough valid cases for processing Egger’s test for attitude outcome, so no statistics were computed (see Fig 3 in S2 Fig).

The trim and fill method suggested that no adjustments were required to correct the asymmetry of the funnel plot. Therefore, any potential publication bias was considered minimal. For physical activity attitude outcomes, there were not enough valid cases for processing thus no statistics were computed (see Fig 3 in S2 Fig).

Discussion:

Limitations- Lastly, given the moderate to high heterogeneity among the experiential learning physical activity interventions, may have distorted our conclusions reached.

Supporting information:

S1 Fig. Meta-analysis output for behaviour outcome.

S2 Fig. Meta-analysis output for attitude outcome.

---

## [Editor Report · Decision Letter 2]

14 Nov 2023

The effect of experiential learning interventions on physical activity outcomes in children: A systematic review

PONE-D-23-02390R2

Dear Dr. Varman,

We’re pleased to inform you that your manuscript has been judged scientifically suitable for publication and will be formally accepted for publication once it meets all outstanding technical requirements.

Kind regards,

Dean Dudley, PhD

Guest Editor

PLOS ONE
---

## [Editor Report · Acceptance letter]

20 Nov 2023

PONE-D-23-02390R2 

The effect of experiential learning interventions on physical activity outcomes in children: A systematic review 

Dear Dr. Varman:

I'm pleased to inform you that your manuscript has been deemed suitable for publication in PLOS ONE. Congratulations! Your manuscript is now with our production department. 

Kind regards, 

on behalf of

Professor Dean Dudley 

Guest Editor

PLOS ONE